# Homeostatic synaptic normalization optimizes learning in network models of neural population codes

Jonathan Mayzel, Elad Schneidman*

Department of Brain Sciences, Weizmann Institute of Science, Rehovot, Israel

## eLife Assessment

This work is an **important** contribution to the development of a biologically plausible theory of statistical modeling of spiking activity. The authors **convincingly** implemented the statistical inference of input likelihood in a simple neural circuit, demonstrating the relationship between synaptic homeostasis, neural representations, and computational accuracy. This work will be of interest to neuroscientists, both theoretical and experimental, who are exploring how statistical computation is implemented in neural networks.

*For correspondence:
elad.schneidman@weizmann.ac.il

Competing interest: The authors declare that no competing interests exist.

**Abstract** Studying and understanding the code of large neural populations hinge on accurate statistical models of population activity. A novel class of models, based on learning to weigh sparse nonlinear Random Projections (RP) of the population, has demonstrated high accuracy, efficiency, and scalability. Importantly, these RP models have a clear and biologically plausible implementation as shallow neural networks. We present a new class of RP models that are learned by optimizing the randomly selected sparse projections themselves. This 'reshaping' of projections is akin to changing synaptic connections in just one layer of the corresponding neural circuit model. We show that Reshaped RP models are more accurate and efficient than the standard RP models in recapitulating the code of tens of cortical neurons from behaving monkeys. Incorporating more biological features and utilizing synaptic normalization in the learning process, results in accurate models that are more efficient. Remarkably, these models exhibit homeostasis in firing rates and total synaptic weights of projection neurons. We further show that these sparse homeostatic reshaped RP models outperform fully connected neural network models. Thus, our new scalable, efficient, and highly accurate population code models are not only biologically plausible but are actually optimized due to their biological features. These findings suggest a dual functional role of synaptic normalization in neural circuits: maintaining spiking and synaptic homeostasis while concurrently optimizing network performance and efficiency in encoding information and learning.

## Introduction

The potential 'vocabulary' of spiking patterns of a population of neurons scales exponentially with the size of the population, and so, mapping the rules of neural population codes and their semantic organization, cannot rely on direct sampling of the vocabulary for more than a handful of neurons. Moreover, the stochastic nature of neural activity implies that the characterization of neural codes must rely on probability distributions over population activity patterns. Therefore, to describe and analyze the structure and content of the code with which neural circuits respond to stimuli, process information, and direct action – we must learn statistical models of their activity. Such models have been used to study neural population codes in different systems: Models of the directional coupling between

neurons, such as Generalized Linear Models, have been used to replicate the stimulus-dependent rates of populations of tens of neurons (*Truccolo et al., 2005*; *Pillow et al., 2008*; *Calabrese et al., 2011*; *Weber et al., 2012*). Maximum entropy models have accurately captured the joint activity patterns of more than 100 neurons, using simple statistical features of the population, like firing rates, pairwise correlations, synchrony, and other low-order statistics (*Schneidman et al., 2006*; *Shlens et al., 2006*; *Tang et al., 2008*; *Tkačik et al., 2014*; *Ganmor et al., 2011*; *Marre et al., 2009*; *Ohiorhenuan et al., 2010*; *Granot-Atedgi et al., 2013*; *Meshulam et al., 2017*). These models have further been used to characterize the semantic organization of population codes (*Ganmor et al., 2015*; *Tkačik et al., 2013a*). Auto-encoder models have been employed to replicate the detailed structure of population activity – yielding generative models that can be used to study the code, but their design is difficult to interpret (*Pandarinath et al., 2018*; *Barrett et al., 2019*; *Gonçalves et al., 2020*). Importantly, scaling of these models to hundreds of neurons is computationally demanding (*Tkačik et al., 2015*; *Meshulam et al., 2019*; *Ganmor et al., 2011*), which has been a major challenge in modeling large neural systems.

While statistical models are invaluable for describing and studying neural codes, it is not clear whether the brain relies on such models or implements them when representing or processing information (*Schneidman, 2016*; *Karpas et al., 2019*). Consequently, much of the analysis of neural codes has focused on decoding population activity, typically using simple decoders (*Panzeri et al., 2017*; *Tkačik et al., 2013b*; *Pillow et al., 2008*; *Botella-Soler et al., 2018*; *Shi et al., 2019*; *Whiteway et al., 2020*), or metrics over the structure of population activity patterns (*Ganmor et al., 2015*; *Gallego et al., 2020*; *Chaudhuri et al., 2019*). Yet, if neural circuits do implement such statistical models, and in particular, ones that compute the likelihood of their inputs – this would present a realizable mechanism for real neural circuits to carry Bayesian computation and decision making (*Maoz et al., 2020*; *Vertes and Sahani, 2018*; *Zemel et al., 1998*). Such network models are, therefore, of interest not only as a way to study neural codes, but also as a potential way for biological neural networks to implement efficient learning and overcome the credit assignment problem. In addition, they may be useful for improving learning in artificial neural networks using biological features (*Bengio et al., 2016*; *Yamins and DiCarlo, 2016*; *Poirazi et al., 2003*; *Richards et al., 2019*; *Zhong et al., 2022*; *Chavlis and Poirazi, 2021*).

Both structured architectural features of neural circuits and random connectivity patterns have been suggested to shape the computation carried out by neural circuits (*Litwin-Kumar et al., 2017*; *Maoz et al., 2020*; *Haber and Schneidman, 2022a*; *Kim et al., 2019*; *Pechuk et al., 2022*; *Haber and Schneidman, 2022b*). These computations rely on the nature of synaptic connectivity and the coupling between synapses in terms of how they change during learning. Competition mechanisms between synapses or other regularization mechanisms have also been suggested to be important components of computation and learning in artificial neural networks as well as in cortical circuits (*Heeger, 1992*; *Carandini and Heeger, 2011*). One such mechanism is the homeostatic scaling of synaptic plasticity, which has been observed in vitro and in vivo at the level of incoming synapses to a neuron and outgoing ones (*Turrigiano et al., 1998*; *Keck et al., 2013*; *Hengen et al., 2013*; *Turrigiano, 2008*). This mechanism has been commonly attributed to the regulation of firing rates, while its functional implications remain mostly unclear, but of interest computationally and mechanistically (*El-Boustani et al., 2018*; *Wu et al., 2020*; *Keck et al., 2017*; *Zenke and Gerstner, 2017*; *Toyoizumi et al., 2014*). A related computational feature has been presented by network models that include divisive normalization, suggested as an important component of computations performed by cortical circuits (*Simoncelli and Heeger, 1998*).

Here, we bring these ideas together to present a biologically-inspired variant of a new family of statistical models for large neural population codes. Adding biological features to these population models enabled us to improve the models, and to explore designs that real neural circuits could employ to implement such models. Specifically, we expand the Random Projections (RP) model (*Maoz et al., 2020*), which was shown to be highly accurate in recapitulating the detailed spiking patterns of more than 100 neurons in different neural systems. Importantly, in addition to being accurate and requiring little amounts of training data, these RP models can be readily implemented by a simple neural circuit model – suggesting how real neural circuits can learn a statistical model of their own inputs and compute the likelihood of the inputs. We show that we can make these models better by 'reshaping' the randomly chosen sparse non-linear projections that they rely on, achieving highly

accurate models using significantly fewer projections. We further show that reshaping of projections that incorporates normalization of synaptic weights during learning, results in more accurate models that are also more efficient, and makes the models homeostatic in terms of neural activity and total synaptic weights. Thus, we present a new class of accurate and efficient statistical models for large neural population codes that also suggests a clear computational benefit of homeostatic synaptic normalization and its potential role in biological neural networks and artificial ones.

## Results

The RP model is a class of highly accurate, scalabale, and efficient statistical models of the joint activity patterns of large populations of neurons (*Maoz et al., 2020*; *Vertes and Sahani, 2018*). These models are based on random and sparse nonlinear functions, or 'projections', of the population: Given a recording of the spiking activity of a population of neurons, the model is a probability distribution over discrete activity patterns (quantized into small time bins, e.g. 10–20ms), that relies on a set of random non-linear functions of the population activity,

$$f_i(\vec{x}) = \sum_j \left( \sum a_{ij} x_j - \theta_i \right),$$ (1)

where $a_{ij}$ are randomly sampled coefficients such that most of them for any $i$ are zero (i.e. the set is sparse), $\theta_i$ are thresholds, and $\sigma(\cdot)$ are nonlinear functions (e.g. the Heaviside step function). The RP model is the maximum entropy distribution $p(\vec{x})$ (*Jaynes, 1957*), which is consistent with the observed

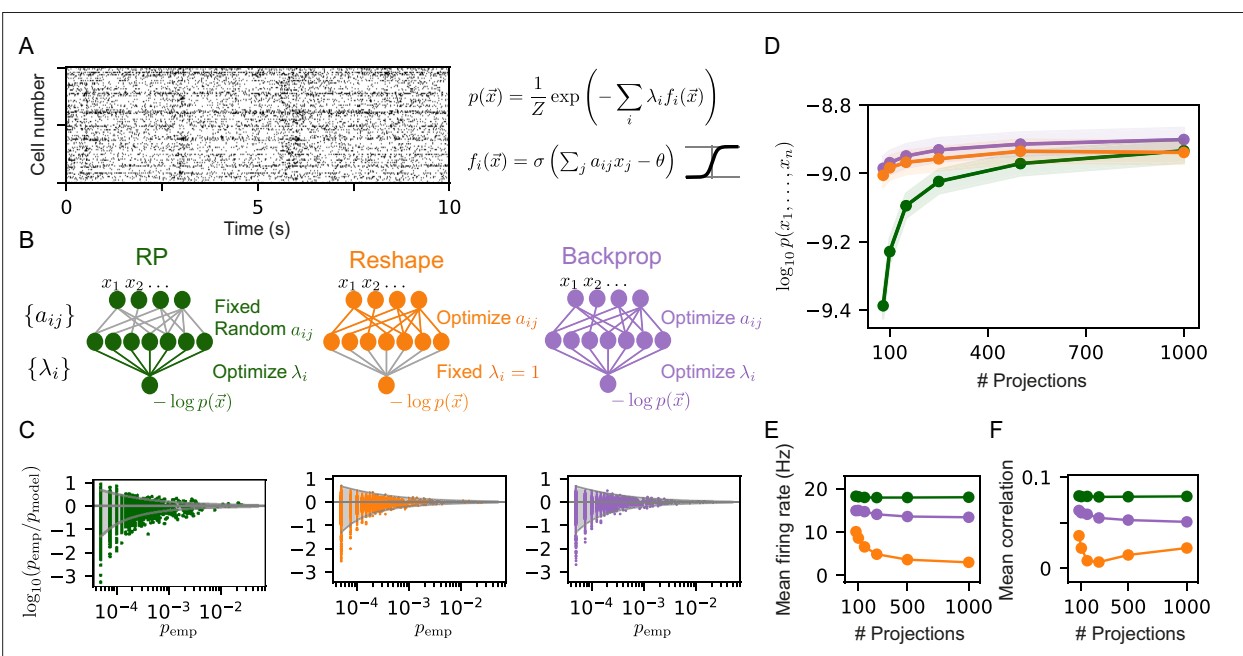

**Figure 1.** Reshaped Random Projections (RP) models outperform RP models. (**A**) A short segment of the spiking activity of 100 cortical neurons used for the analysis and comparison of different statistical models of population activity (see Materials and methods). (**B**) Schematics of the neural circuits that implement the different RP models we compared: The 'standard' RP model, where the coefficients $a_{ij}$ that define the projections are randomly selected and fixed whereas the factors $\lambda_i$ are learned (see text). The Reshaped RP model, where the coefficients $a_{ij}$ that define the projections are tuned and the factors $\lambda_i$ s are fixed. The backpropagation model, where we tune both the $a_{ij}$ s and $\lambda_i$ s. (**C**) The predicted probability of individual populations' activity patterns as a function of the observed probability for RP, Reshaped, and backpropagation models. Gray funnels denote 99% confidence interval. (**D**) Average performance of models of the three classes is shown as a function of the number of projections, measured by log-likelihood, over 100 sets of randomly selected groups of 50 neurons. Reshaped models outperform RP models and are on par with backpropagation; the shaded area denotes the standard error over 100 models. (**E–F**) Mean firing rates of projection neurons and mean correlation between projections. Reshaped models show lower correlations and lower firing rates compared to RP and backpropagation models. Standard errors are smaller than marker's size, hence invisible.

The online version of this article includes the following figure supplement(s) for figure 1:

**Figure supplement 1.** Comparison of different Random Projections (RP) model variants .

**Figure supplement 2.** Comparison of different Reshaped Random Projections (RP) model variants.

average values of the random projections, $\langle f_i \rangle_p = \langle f_i \rangle_{data}$ (see Materials and methods). Thus, it is the least structured distribution that retains the average values of the projections, is mathematically unique, and is given by

$$p_{RP}(\vec{x}) = \frac{1}{Z} \exp\left(-\sum_i \lambda_i f_i(\vec{x})\right),$$ (2)

where $\lambda_i$ are Lagrange multipliers, and $Z$ is a normalization factor or the 'partition function', which can be found numerically. Applied to cortical data from multiple areas (see, e.g. *Figure 1A*), this model proved to be highly accurate in predicting individual activity patterns, using small amounts of training data (*Maoz et al., 2020*). Importantly, unlike many other statistical models of population activity, RP models have a simple, biologically plausible neural circuit that can implement them (*Maoz et al., 2020*): *Figure 1B* shows such a feed-forward circuit with one intermediate layer and an output neuron, where the random coefficients of the sparse projections, $a_{ij}$, are the synaptic weights connecting the input neurons $\vec{x}$ to an intermediate layer of neurons $\{f_i\}$. Each intermediate neuron implements one projection of the input population. The Lagrange multipliers, $\lambda_i$, are the synaptic weights connecting the intermediate layer to the output neuron, whose membrane potential or output gives the log-likelihood of the activity pattern of $\vec{x}$, up to a normalization factor.

The model in *Equation 2* harbors a duality between the projections, $f_i$, and their coefficients, $\lambda_i$: In the maximum entropy formalism of the model, the projections are randomly sampled and then fixed, and their corresponding weights, $\lambda_i$'s, are tuned to maximize the entropy and satisfy the constraints. Alternatively, we may consider the case of training the model by keeping the $\lambda_i$'s fixed and changing or tuning the projections $f_i$ to maximize the likelihood. In the corresponding neural circuit, this would imply that we would learn a circuit that implements the statistical model by training the sparse set of synaptic connections, $a_{ij}$, which define the projections, instead of training the synapses that weigh the projections, $\lambda_i$ (*Figure 1B*).

Notably, a variant of the RP model in which projections that were weighted by a low value of $\lambda_i$ are pruned and replaced with new RP proved to be more accurate than the original RP model, while using fewer projections (*Maoz et al., 2020*). This procedure of pruning and replacement is a crude form of learning of the model through changing the projections, and finding more efficient ones. We, therefore, asked here whether instead of the heuristic pruning and replacement, we can directly learn more accurate and efficient models by tuning the projections.

## Reshaping RP gives more accurate and compact models

We first learned a new class of RP models for populations of tens of cortical neurons from the prefrontal cortex of monkeys performing a visual classification task (*Kiani et al., 2014*) by tuning their randomly selected projections. Specifically, given an initial draw of sparse projections, the random weights that define the projections, $a_{ij}$, are then changed to maximize the likelihood of the model:

$$\Delta a_{ij} = \eta \lambda_i \left( \left\langle \frac{\partial \sigma(\vec{x})}{\partial a_{ij}} \right\rangle_p - \left\langle \frac{\partial \sigma(\vec{x})}{\partial a_{ij}} \right\rangle_{data} \right),$$ (3)

where $\eta$ is the learning rate. We note that unlike the RP model presented in *Maoz et al., 2020*, here we used a sigmoid function for the nonlinearity of the projections,

$$\sigma(x) = \frac{1}{1 + e^{-\beta x}},$$ (4)

where $\beta$ sets the slope of the sigmoid. In this formulation, the model ranges from an independent model of the population for $\beta \to 0$, to the original RP model of *Maoz et al., 2020* for $\beta \to \infty$. The rule for changing the projections (*Equation 3*) means that the specific set of inputs to each projection neuron is retained, but their relative weights are changed, and so the projections are 'reshaped'.

We compared the RP and the Reshaped RP models by quantifying their performance on the same set of initial projections. We first learned the RP model as in *Maoz et al., 2020*, using a Heaviside non-linearity for the projections, and RP models that used a sigmoid non-linearity, where both models used the same set of RP, and found the latter models to be be more accurate (see *Figure 1—figure supplement 1A*). We then learned Reshaped RP models in which we optimize the same initial projections

while keeping all $\lambda_i = 1$. We note that while in its maximum entropy formulation, the RP model is the unique solution to a convex optimization problem, the Reshaped RP models are not guaranteed to reach a global optimum. We also considered another class of models, in which the projections and the Lagrange multipliers $\lambda_i$ are optimized simultaneously, similar to backpropagation-based learning used to train feed-forward neural networks (see Materials and methods). *Figure 1C* shows an example of the accuracy of the sigmoid RP models, Reshaped RP models, and backpropagation-based models in predicting the probability of individual activity patterns for one group of 20 neurons, recorded from the cortex of behaving monkeys (*Kiani et al., 2014*). The activity patterns are predicted by the reshaped RP model to an accuracy that is within the sampling noise (denoted by the 99% confidence interval funnel), and is similar to the performance of the full backpropagation model. The standard RP model, in comparison, has many more patterns that are outside the 99% confidence interval funnel. We quantified the performance of the three classes of models by calculating the mean log-likelihood of the models over 100 groups of 50 neurons on held out datasets, as a function of the number of projections that we used (*Figure 1D*). The reshaped models outperform the RP ones for a low number of projections, whereas the performances of all three models converge to a similar value for large number of projections.

Because reshaping may change all the existing synapses of each projection, the number of parameters is the number of projections times the projections in-degree. While this is much larger than the number of parameters that we learn for the RP model (one for each projection), we submit that the performance of the reshaped models is not a mere result of having more parameters. In particular, we have seen that RP models that use a small set of projections can be very accurate when the projections are optimized using the pruning and replacement process (*Maoz et al., 2020*; see also *Figure 1— figure supplement 1B*). Thus, it is really the nature of the projections that shapes the performance. Indeed, our results here show that a small fixed connectivity projection set with weight tuning is enough for accurate performance which is on par or better than an RP model with more projections.

To compare the 'mechanistic' nature of these different models, we calculated the mean correlation between the projections within each model class, and the average values of each projection (where the average is over the population activity patterns), which correspond to the mean firing rates of the neurons in the intermediate layer. Interestingly, the firing rates of the neurons in the intermediate layer are considerably lower for the reshaped models, and this sparseness in activity becomes more pronounced as a function of the number of projections (*Figure 1E*). We further find that the correlations between the projections in the reshaped models are considerably lower compared to RP and backpropagation models (*Figure 1F*).

The projections' thresholds $\theta_i$, which are analogous to the spiking thresholds of the projection neurons, may affect the performance of the models. We, therefore, asked how optimizing $\theta_i$, in addition to reshaping the coefficients of each projection, affect the reshaped RP and the backpropagation models. We find that this addition has a small effect on the performance of the models in terms of their likelihood (*Figure 1—figure supplement 2A*). We also find that this has a small effect on the firing rates of the projection neurons: backpropagation models with tuned thresholds show lower firing rates compared to backpropagation models with fixed threshold, whereas reshaped RP models with optimized thresholds show higher firing rates compared to models with fixed threshold. Yet, both versions of the reshaped RP models show lower firing rates compared to both versions of the backpropagation models. Given the small effect of tuning threshold on models' performance and their internal properties, we henceforth focus on Reshaped RP models with fixed thresholds.

An additional set of parameters that might affect the Reshaped RP models are the coefficients $\lambda_i$, that weigh each of the projections. Above, we used $\lambda_i = 1$ for all projections, here we investigated the effect of the value of $\lambda$ on the performance of the Reshaped RP models (*Figure 1—figure supplement 2B*). We find that for models with a small set of projections, high values of $\lambda$ result in better performance than models with low values. We find an opposite relation for models with large number of projections. (We submit that the performance decrease of Reshaped RP models with high value of $\lambda$, as the number of projections grows, is a reflection of the non-convex nature of the Reshaped RP optimization problem). The mean firing rates of the projection neurons for models with different values of $\lambda$ show a clear trend, where higher values result in lower mean firing rates. Thus, we conclude that there is an interplay between the number of projections and the value of $\lambda$ one should pick. For the population sizes and projection sets we have used here, $\lambda = 1$ is a good choice,

but, we note that in general, one should seek the appropriate value of $\lambda$ for different population sizes or data sets.

Thus, the reshaped projection models suggest a way to learn more accurate models of population activity, by tuning of projections. These models are also more efficient, requiring fewer projections. These projections also have lower firing rates (i.e. reshaped projections use fewer spikes), and they are less correlated. Given their accuracy and efficiency, we next asked how adding biological features or constraints to a Reshaped RP circuit may affect its performance and efficiency.

## Normalized reshaping of RP gives more accurate and efficient models

We studied the effect of adding two classes of biological features or constraints on the performance and nature of the Reshaped RP circuit model. The first constraint stems from the biophysical limits on individual synapses, and so we bound the maximal strength of individual synapses such that the strength of all synaptic weights are smaller than a 'ceiling' value: $|a_{ij}| < \omega$. The other is a normalization of the synaptic weights during the reshaping, inspired by the synaptic re-scaling that has been observed experimentally (*Turrigiano, 2008*), and divisive normalization of synaptic weights (*Heeger, 1992*). We consider multiple mechanisms of this kind later, but begin here with fixing the total sum of the incoming synaptic strength of each projection such that $\sum_j |a_{ij}| = \phi$. Thus, when the strength of one synapse increases (decreases), the strength of the rest of the incoming synapses decreases (increases) such that the total synaptic weight incoming into the projection is kept constant. We term this constraint 'homeostatic synaptic normalization'. We emphasize that the notion of homeostatic mechanisms is commonly reserved for designating regulation processes that retain a functional property of neurons, whereas normalization of synaptic weights might seem more mechanistic than functional. But, as we show later, learning with synaptic normalization also regulates the firing rate of the projection neurons, and so, we use this name henceforth.

To compare the effect of these constraints, we used the same set of initial RP, and then learn by reshaping them, each time with a different value of their corresponding parameters, $\phi$ or $\omega$. We estimated the likelihood of each of the models on 100 groups of 50 neurons, over 100 random sets of 150 projections. To quantify the 'synaptic budget' of each model, we measured the total sum of the absolute values of synaptic weights available to each model in units of the total synaptic strength of the initial set of projections (this is equivalent to defining the total sum of the synaptic weights of the initial set of projections as '1', and then measuring total synaptic weights in these units). For the models with bounded synapses, the total available synaptic budget is given by the number of synapses times $\omega$, whereas for the homeostatic constraint, it equals $\phi$ times the number of projections in the model. *Figure 2B* shows the log-likelihood of each model class vs. the total available synaptic budget of the different models: For a wide range of synaptic budgets, the homeostatic models outperform the bounded models, and only for very high values of available synaptic budget, the performance of the bounded models is on par with the homeostatic models.

The differences between the homeostatic normalization models and the bounded synaptic strength models are further reflected in *Figure 2C*, which shows the performance of each model class as a function of the total sum of synaptic weights that is used by that model at the end of the training, $\sum_{ij} |a_{ij}|$. We note that the curve of the homeostatic model is identical to the one from *Figure 2B* by definition; the curve of the bounded models shows that at a certain value of $\omega$ the sum of the synaptic weights starts to decrease and converges to the unconstrained reshaped model. The poor performance of the bounded models compared to the homeostatic ones suggests that the coupled changes in the synaptic weights improve learning. Specifically, during reshaping, the homeostatic models move synaptic "mass" from less important synapses to more important ones. This redistribution of resources results in accurate models even for relatively low values of synaptic weights – making them more efficient in terms of the total synaptic weight needed.

The dominance of the homeostatic learning over the bounded synaptic weights is clear not just for the average over models, but also at the level of individual models: *Figure 2D* shows the performance of the homeostatic and bounded models that are initialized with the same set of RP; all bounded constraint models are inferior to the corresponding RP ones, whereas all the homeostatic constraint models are superior to the RP models (and clearly all the homeostatic models are superior to the corresponding bounded models).

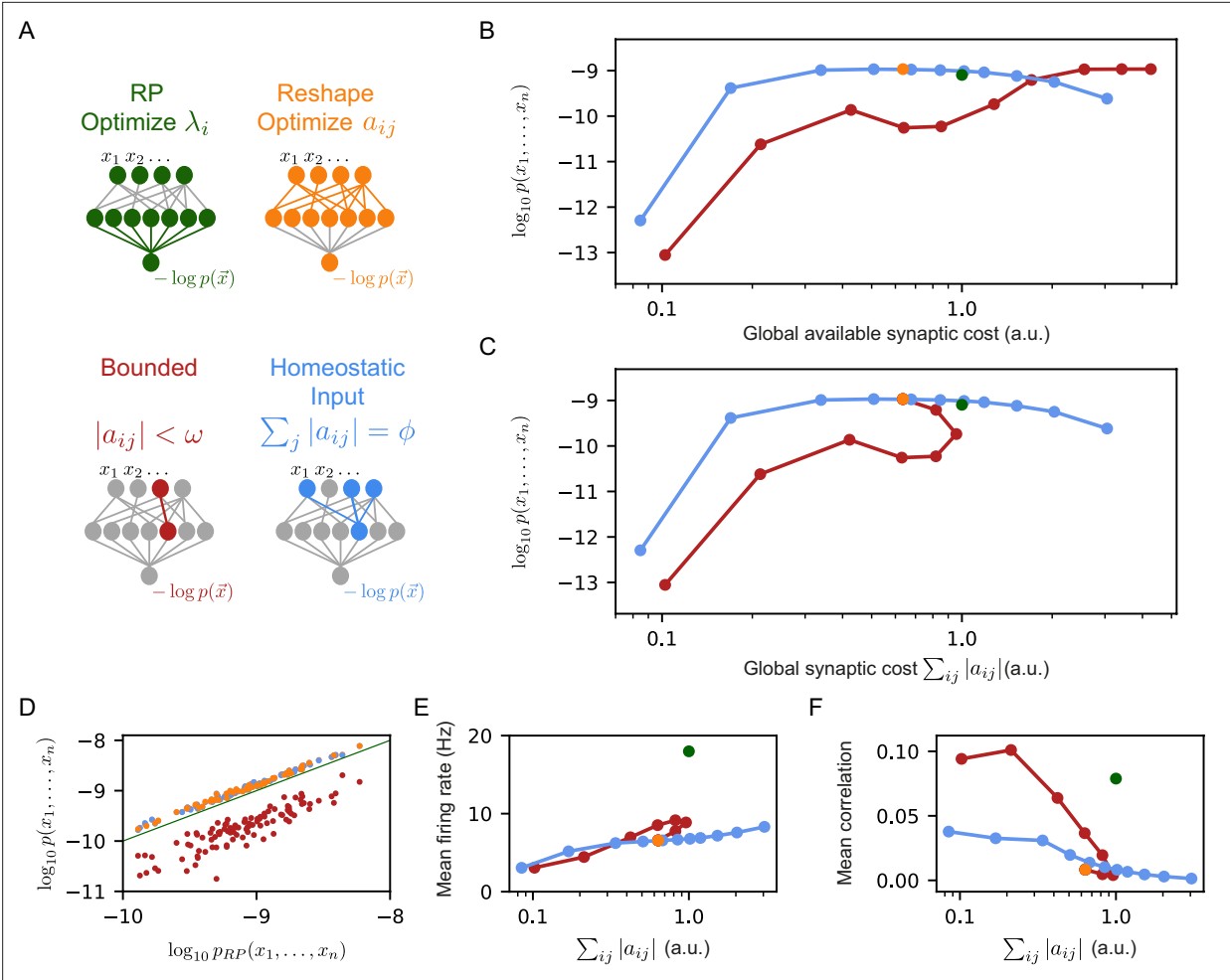

**Figure 2.** Reshaped RP models that use homeostatic synaptic normalization outperform RP models and bounded RP models. (**A**) Schematic drawing of the different models we studied: standard RP model, unconstrained reshaped model, and two types of constrained reshaped models: Bounded models in which each synapse separately obeys $|a_{ij}| < \omega$ during learning, and normalized input reshaped models, where we fix the total synaptic weight of incoming synapses $\sum_j |a_{ij}| = \phi$. (**B**) The mean log-likelihood of the models is shown as a function of the total available budget. The normalized input reshaped RP models give optimal results for a wide range of values of available synaptic budget, outperforming the bounded models and the RP model. (**C**) The mean log-likelihood of the models is shown as a function of the total used budget. Aside from the bounded models, all other models are the same as in (**B**) by construction. For high available budget values, bounded models show better performance while utilizing a lower synaptic budget, similar to the unconstrained reshape model. (**D**) Comparison of the performance of 100 individual examples of each model class and their corresponding RP models, where all models relied on the same set of initial projections. Normalized input models outperformed the RP models in all cases (all points are above the diagonal), while all bounded models were worse (points below the diagonal). (**E–F**) The mean correlation and firing rates of projections as a function of the model's cost. Normalized reshape models show low correlations and mean firing rates, similar to unconstrained reshaped models. Note that in panels B, C, E, and F, the standard errors are smaller than the marker size, and are therefore invisible.

The online version of this article includes the following figure supplement(s) for figure 2:

**Figure supplement 1.** Homeostatic models are superior to Reshaped RP models for every choice of $\lambda$.

We further find that the mean firing rates of the reshaped projection neurons, as well as the correlations between them, are lower in the homeostatic models compared to the bounded models (*Figure 2E–F*), making them more energetically efficient (in terms of spiking activity). We recall that this is consistent with the notions of efficient coding by decorrelated neural populations (*Barlow, 1961*; *Olshausen and Field, 1997*).

Exploring the effect of synaptic normalization on models with different values of $\lambda$ (*Figure 2— figure supplement 1*), we find that homeostatic Reshaped RP models are superior to the non-homeostatic Reshaped RP models: For low values of $\lambda$, the homeostatic and Reshaped RP models show similar performance in terms of log-likelihood, whereas the homeostatic models are more

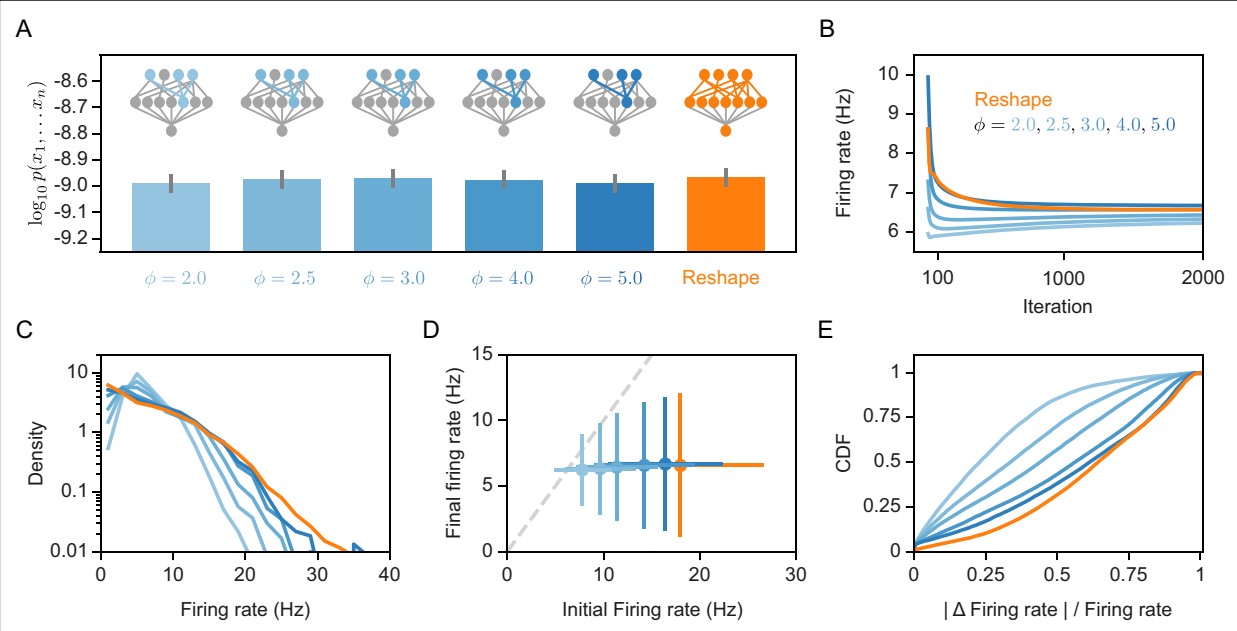

**Figure 3.** Homeostatic models' performance and firing rates. (**A**) Likelihood values of the different homeostatic models we consider here; Error bars denote standard error over 100 different models. (**B**) Projections' mean firing rates during reshaping; Standard error is smaller than marker size, hence cannot be seen (**C**) A histogram of the projections' firing rates of unconstrained models and different homeostatic input models. (**D**) Projections' mean firing rates after reshaping vs. the projection's initial firing rate, for different homeostatic models and unconstrained model; Error bars denote standard deviation over 100 different models. (**E**) The cumulative density function of the projections' firing rate relative change during reshape, $\left|FR_{final} - FR_{initial}\right|/FR_{initial}$.

efficient. Importantly, for high values of $\lambda$ homeostatic models are not only more efficient but also show better performance. We conclude that the benefit of the homeostatic model is insensitive to the specific choice of $\lambda$.

## Normalized reshaping of RP results in more efficient codes and homeostasis of firing rates

The experimental characterization of synaptic re-scaling has shown it to be a homeostatic mechanism that regulates the firing rates of neurons (*Turrigiano, 2008*). We therefore asked whether the synaptic normalization we employ for the Reshaped RP models has a similar effect. *Figure 3A* shows that the overall performance of the model in terms of capturing the population codebook is similar between the 'free' reshape model and different values of synaptic normalization. Similarly, reshaping with normalization or without it drives the projection neurons to converge to similar average firing rate values (*Figure 3B*). However, the distribution of firing rates over the different neurons becomes narrower with tighter normalization values (*Figure 3C*). Importantly, while different normalization values imply very different initial firing rates of the projection neurons, after reshaping the values converge to similar average values (*Figure 3D*). Moreover, reshaping with normalization implies smaller changes in the reshaping process (*Figure 3E*). Thus, normalized reshaping results in homeostatic regulation of the firing rates, which validates the naming of these models as homeostatic normalization reshaping of RP.

Having established the computational benefits and efficiency of the homeostatic reshaped projection models that rely on synaptic normalization, we turned to ask how the connectivity itself, rather than the synaptic weights, may affect the performance of the models.

## Optimal sparseness of Reshaped Projections models under homeostatic constraints

The benefits of reshaping a given set of projections, reflected in the figures above, raise the question of the importance of the nature of the RP we choose (which are then reshaped). We, therefore, asked

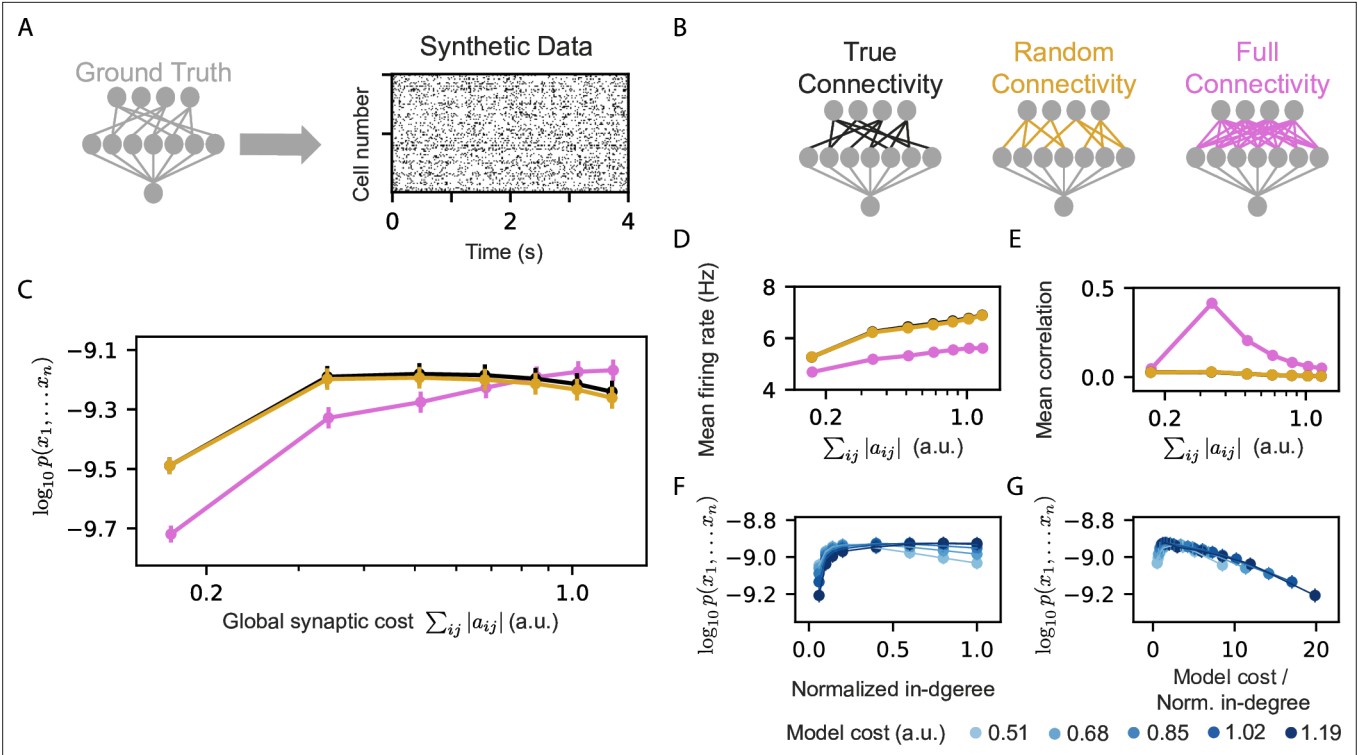

**Figure 4.** Models that rely on projections that use random connectivity show similar performance to models that use the correct connectivity.
(**A**) Synthetic population activity data is sampled from an RP model with known connectivity (i.e. the 'ground truth' model; see Materials and methods).
(**B**) Homeostatic Reshaped random projections models that differ in their connectivity are learned to fit the synthetic data. The 'True connectivity' model uses projections whose connectivity is identical to the 'ground truth' model. The 'Random connectivity' model uses projections that are randomly sampled using sparse random connectivity. The 'Full connectivity' model is a homeostatic reshaped model that uses projections with full connectivity.
(**C**) The mean log-likelihood of the models is shown as a function of the model's cost. The true connectivity model is only slightly better than the random connectivity model, with both outperforming the full connectivity model for low model budget values. (**D**) The firing rates of the projection neurons, shown as a function of the model cost. (**E**) The mean correlation between the activity of the projection neurons, shown as a function of the model cost. We note that true and random connectivity models are indistinguishable. (**F**) The performance of homeostatic reshaped RP models, shown as a function of their normalized in-degree of the projections (0–disconnected, 1–fully connected), for different normalization values, shown by the model's synaptic cost. (**G**) The performance of the homeostatic Reshaped RP models, shown as a function of the synaptic cost normalized by the in-degree of the projections. Curves of different cost values coincide, suggesting a fixed optimal cost/activity ratio. Note that in panels D-G the standard errors over 100 models are smaller than the size of markers and are, therefore, invisible.

how the initial random 'wiring' of the projections affects the performance of the model, and whether non-RP would result in even better models. To quantify the effects of the projections' connectivity on the performance and efficiency of reshaped models, we used simulated population activity that we generated using RP models that were trained on real data. By using synthetic data that was generated by a known model, we can compare the learned models to the 'ground truth' in terms of connectivity, as well as extensively sampling of activity patterns from the model.

We learned homeostatic reshaped models for the synthetic data, using different initial connectivity structures (*Figure 4A–B*): (i) A 'true' connectivity model in which we reshaped a RP model that has the same connectivity as the projections of the model that generated the data. (ii) A Random connectivity model in which we reshaped projections with sparse and connectivity that is randomly sampled and is independent of the model that generated the synthetic data. (iii) A full connectivity model in which we reshaped RP with full connectivity, that is all input neurons are connected to all the projections, but with random initial weights. We carried out homeostatic reshaping of the projections in all three models with different values of $\phi$. Surprisingly, the true and random connectivity models performed very similarly (*Figure 4C*). Although the full connectivity model contains the 'ground truth' connectivity, and could recreate the true connectivity by canceling out unnecessary synapses during reshaping – we find that the full connectivity models are inferior to the other models, except for the case of high model costs.

The mean correlations between projections at the end of reshaping and the mean firing rates of the models that use the true and random connectivity were also very similar (*Figure 4D–E*), whereas the full connectivity models showed, again, very different behavior. These results reflect another computational benefit of homeostatic reshaping: there is no need to know the optimal circuit connectivity, and there is no apparent benefit to all-to-all connectivity, which would be expensive in terms of the energetic cost, the space needed, and the biological construction. Thus, starting from random connectivity and optimizing the circuit under homeostatic constraints seems to provide optimal results.

Given the inefficiency of the fully connected reshaped projections model, we also quantified the effect of the sparseness of the projections on reshaped RP models. We recall that for the standard RP model, sparse projections were optimal for a wide range of network sizes (*Maoz et al., 2020*), and so we measured the performance of homeostatic reshaped RP models for different values of in-degree of the projections, while keeping the total synaptic budget of the models fixed. We found that different synaptic budgets have a different optimal in-degree (*Figure 4F*), and that the value of the optimal in-degree seems to grow with the total synaptic budget.

We further estimated the efficiency of the models by the synaptic cost per connection in the projections (*Figure 4G*). We find that curves for different total synaptic costs seem to coincide and have a similar peak value – suggesting an optimal ratio between the total available resources and the number of synapses.

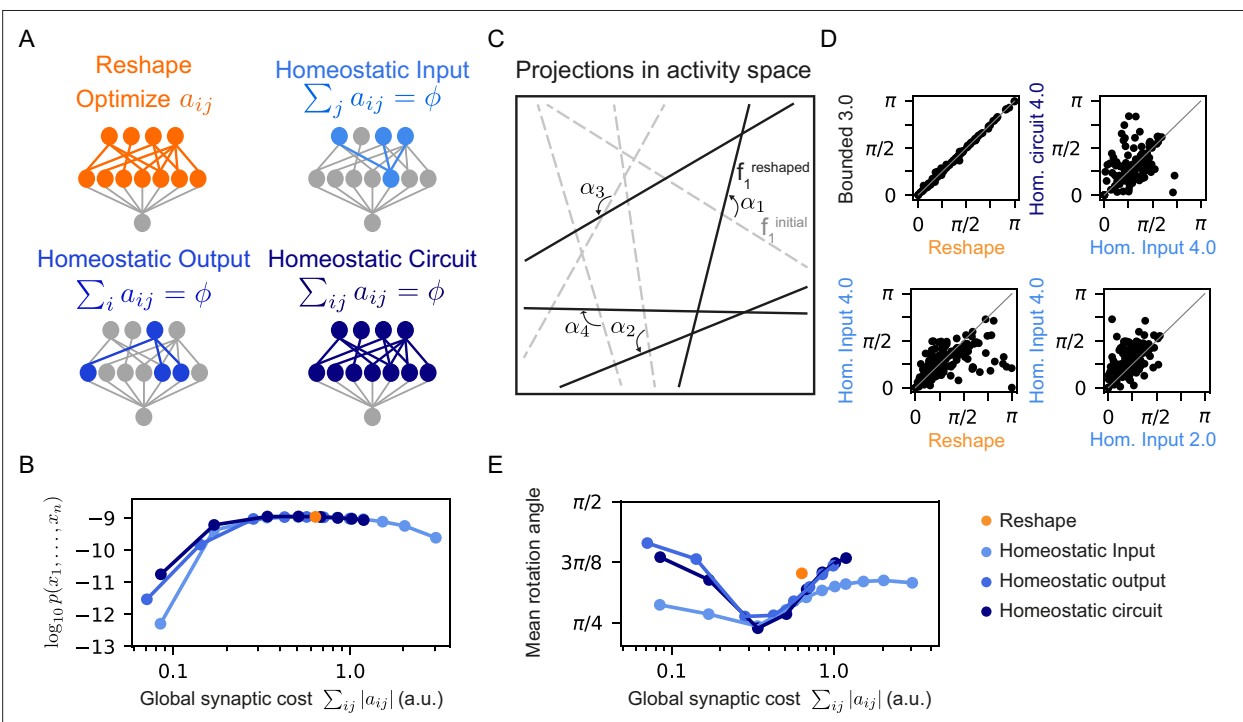

**Figure 5.** Different homeostatic synaptic normalization mechanisms result in similar model behavior, for a wide range of random projection parameters. (**A**) Schematic drawings of the different homeostatic models we compared: Homeostatic input models in which we fixed the total synaptic weight of the incoming synapses $\sum_j |a_{ij}| = \phi$; Homeostatic output models in which we fixed the total synaptic weight of the outgoing synapses $\sum_i |a_{ij}| = \phi$; and Homeostatic circuit models in which we fixed the total synaptic weight of the whole synaptic circuit $\sum_{ij} |a_{ij}| = \phi$. (**B**) The mean log-likelihood of models, shown as a function of the total used synaptic cost. All three homeostatic model variants show similar behavior. (**C**) Schematic drawing of how projections rotate during reshaping: starting from the initial projections (grey lines), they rotate to their reshaped orientation (black lines) by angle $\alpha_i$. (**D**) Rotation angles after reshaping, shown for different pairs of models. All four panels show models that initialized with the same set of projections. The different labels specify the constraint type and strength, namely, the specific value of $\phi$ and $\omega$. (**E**) The mean rotation angle of the projections due to reshaping, shown as a function of the model synaptic budget.

The online version of this article includes the following figure supplement(s) for figure 5:

**Figure supplement 1.** Homeostatic model variants show similar firing rates and correlation, but different rotation angles and distribution of weights.

## Different homeostatic mechanisms for reshaping RP models result in different projection sets

We explored two other forms of synaptic normalization rules for the reshaping of projections (*Figure 5A*). In the first, we fixed and normalized the outgoing synapses from each neuron, such that $\sum_i |a_{ij}| = \phi$. In the second, we kept the total synaptic weight of the whole circuit fixed, namely, $\sum_{ij} |a_{ij}| = \phi$. *Figure 5B* shows that the performance of the models that use these other homeostatic mechanisms is surprisingly similar in terms of the model's likelihood over the test data, as well as the firing rates of the projection neurons (*Figure 5—figure supplement 1A*), and correlations between them (*Figure 5—figure supplement 1B*).

As the homeostatic reshaping of RP proved to be similarly accurate and efficient for the three homeostatic model variants, we asked which features of normalized reshaping might differentiate between homeostatic models in terms of their performance. Since each projection defines a hyperplane in the space of population activity patterns, reshaping can be interpreted as a rotation or a change of the angle of these hyperplanes, depicted schematically in *Figure 5C*. We, therefore, compared the different homeostatic variants of the reshaped projections models by initializing them from the same set of RP, and evaluating the corresponding rotation angles, $\alpha$, of all of the projections due to the reshaping.

*Figure 5D* shows an example of the rotations of the same initial projections for one model under different reshaping constraints. While the rotation angles of the bounded model with a high value of $\omega$ is almost identical to the rotation angels of the unconstrained Reshaped RP model (*Figure 5D* top left), as one would expect, the other three panels in 5D reflect substantial differences between models reshaped under different conditions: unconstrained Reshaped RP model vs. a homeostatic one (bottom left), different homeostatic model variants with the same synaptic cost (top right), and homeostatic models with different synaptic cost (bottom right).

*Figure 5E* shows the mean rotation angle over 100 homeostatic models as a function of synaptic cost – reflecting that the different forms of homeostatic regulation results in different reshaped projections. We show in *Figure 5—figure supplement 1C* the histogram of the rotation angles of several different homeostatic models, as well as the unconstrained Reshape model. Interestingly, although the three homeostatic variants show unique rotation angle histograms, they all show a similar minimal mean rotation angle at the same value of synaptic cost. We note that while there is dependency or even redundancy between these different homeostatic mechanisms, it is not immediately clear why their minimal values would be so similar. Analyzing the distribution of the synaptic weights $a_{ij}$ after learning leads to a similar conclusion (*Figure 5—figure supplement 1D*): The peak of the histograms is at $a_{ij} = 0$, implying that during reshaping most synapses are effectively pruned. While the distribution is broader for models with higher synaptic budget, it is asymmetric, showing local maxima at different values of $a_{ij}$.

The diversity of solutions that the different model classes and parameters show imply a form of redundancy in model choice or learning procedure. This reflects a multiplicity of ways to learn or optimize such networks, that biology could use to shape or tune neural population codes.

## Discussion

We presented a new family of statistical models for large neural populations that is based on sparse and random non-linear projections of the population, which are adapted during learning. This new family of models proved to be more accurate than the highly accurate RP class of models, using fewer projections and incurring a lower 'synaptic cost' in terms of the total sum of synaptic weights of the model. Moreover, we found that reshaping of the projections gave even more accurate and efficient models in terms of synaptic weights of the neural circuit that implements the model, and was optimal for random and sparse initial connectivity, surpassing fully connected network models. The synaptic normalization mechanism resulted in homeostatic regulation of the firing rates of neurons in the model.

Our results suggest a computational role for the experimentally observed scaling or normalization of synapses during learning: In addition to 'regularizing' the firing rates in neural circuits, in our Reshaped RP models, homeostatic plasticity optimizes the efficiency of network models in scenarios of limited resources and random connectivity. Moreover, the similarity of the performance of models

that use different homeostatic synaptic mechanisms suggests a possible universal role for homeostatic mechanisms in computation.

We note that while homeostatic synaptic scaling regulates the firing rates of neurons (*Turrigiano, 2008*), it is not immediately clear what 'sets' the desired firing rate of each neuron. The synaptic normalization constraints we used here offer a simple solution: a universal value of the total incoming synaptic weights for the neurons in the circuit (or outgoing ones), results in a widely distributed firing rates of neurons (which may change considerably during the learning), but converge to a similar average value. Thus, rather than requiring some mechanism to define and balance the firing rates of individual neurons, our model suggest a single global synaptic feature that would set this for the RP.

The shallowness of the circuit implementation of the Reshaped RP model implies that the learning of these models does not require the backpropagation of information over many layers, which distinguishes deep artificial networks from biological ones. Moreover, the locality of the reshaping process itself points to the feasibility of this model in terms of real biological circuits. The biological plausibility is further supported by the robustness of the model to the specific connectivity used for the reshaped models, and to the specific choice of the homeostatic mechanism we used.

A key remaining issue for the biological feasibility of the RP family of models is the feedback signal from the readout neuron to the intermediate neurons. The noise-dependent learning mechanism for RP models presented in *Maoz et al., 2020* and for other local feedback and synaptic learning mechanisms that approximate backprogapation (*Poirazi et al., 2003*) offers clear directions for future study. Our results may also be relevant for learning in artificial neural networks, whose training relies on non-convex approaches that necessitate different regularization techniques (*Goodfellow et al., 2016*). The homeostatic mechanism we focused on here is a form of 'hard' L1 regularization, but on the sum of the weights. This approach limits the search space, compared to regularization over the weights themselves, but defines coupled changes in weights, in a manner highly effective for the cortical data we studied. We, therefore, hypothesize that homeostatic normalization may be beneficial for artificial architectures (see, e.g. *Zhong et al., 2022*).

## Materials and methods

### Experimental data

Extra-cellular recordings were performed using Utah arrays from populations of neurons in the prefrontal cortex of macaque monkeys performing a direction discrimination task with random dots. For more details see *Kiani et al., 2014*.

### Data pre-processing

Neural activity was discretized using 20ms bins, such that in each time bin a neuron was active ('1') if it emitted a spike in that bin and silent ('0') if not. Recorded data was split randomly into training sets and held-out test sets: 100 different random splits were generated for each model setup, consisting of 160,000 samples in the training set and 40,000 in the test set.

### Constructing sparse RP

Following (*Maoz et al., 2020*), the coefficients $a_{ij}$ of the RP are set using a two stage process. First, the connectivity of the projections is set such that the average in-degree (*indegree*) of the projections matches a predetermined sparsity value: each input neuron connects to each projection with a probability $p = indegree/n$, where $n$ is the number of neurons in the input layer. The corresponding $a_{ij}$ coefficients are then sampled from a Gaussian distribution, $a_{ij} \sim \mathcal{N}(1, 1)$, and the remaining $a_{ij}$ values are set to zero. The threshold of each projection, $\theta_i$, was set to 1.

The average in-degree of sparse models used here was 5, unless specified otherwise in the text. For the fully connected models *indegree* = $n$ (i.e. sparsity =0).

### Training RP models

Given empirical data $\mathbf{X}$ and a set of projections defined by $a_{ij}$, we train the RP models by searching for the parameters $\lambda_i$ that maximize the log-likelihood of the model given the data, $\arg\max_{\lambda_i}(L(\mathbf{X}))$, where $L(\mathbf{X}) = \sum_{\vec{x} \in \mathbf{X}} \log p_{RP}(\vec{x})$. This is a convex function whose gradient is given by

$$\nabla_{\lambda_i} L(\mathbf{X}) = \langle f_i \rangle_{\mathbf{X}} - \langle f_i \rangle_{p_{RP}} . \tag{5}$$

We found the values $\lambda_i$ that maximize the log-likelihood by gradient descent with momentum or ADAM algorithms. We computed the empirical expectation in $\langle f_i \rangle_{\mathbf{X}}$ by summing over the training data, and the expectation over the probability model $\langle f_i \rangle_{p_{RP}}$ by summing over synthetic data generated from $p_{RP}$ using Metropolis–Hasting sampling.

For each of the empirical marginals $\langle f_i \rangle_{\mathbf{X}}$, we used the Clopper–Pearson method to estimate the distribution of possible values for the real marginal given the empirical observation. We set the convergence threshold of the numerical solver such that each of the marginals in the model distribution falls within a Confidence Interval of one Standard Deviation under this distribution, from its empirical marginal.

### Reshaping RP models

Given empirical data $\mathbf{X}$, we optimize the RP models by modifying the coefficients $a_{ij}$ such that the log-likelihood of the model is maximized, $\arg\max_{a_{ij}}(L(\mathbf{X}))$. Starting from an initial set of projections, $a_{ij}^0$, using the update rule of *Equation 3*, we optimize the projections by applying the gradient descent with momentum algorithm. Importantly, only non-zero elements of $a_{ij}^0$ are optimized.

### Optimizing backpropagation models

Full backpropagation models are optimized using the learning rules of the trained RP models and the reshaped models simultaneously in each gradient descent step, that is *Equations 3 and 5*.

### Homeostatic reshaping of RP models

The homeostatic RP models are reshaped as follows: We first define a set of unconstrained projections where the coefficients $\tilde{a}_{ij}$ are randomly sampled. Each of the projections is then normalized homeostatically, such that $a_{ij}$ are a function of this unconstrained set: $a_{ij} = \phi \cdot \tilde{a}_{ij} / \sum_k |\tilde{a}_{ik}|$, where $\phi$ is the available synaptic budget for each projection. We then optimize $\tilde{a}_{ij}$ to maximize the log-likelihood of the model given the empirical data $\mathbf{X}$: $\arg\max_{\tilde{a}_{ij}}(L(\mathbf{X}))$. The computed constrained projections $a_{ij}$ are then used in the resulting homeostatic RP model.

### Bounded reshaping of RP models

Similar to reshaping homeostatic RP models, we define a set of unconstrained projections $\tilde{a}_{ij}$, where the projections are a function of this unconstrained set: $a_{ij} = \min\left(\max\left(\tilde{a}_{ij}, -\omega\right), \omega\right)$, where $\omega$ is the 'ceiling' value of each synapse.

### Generating synthetic data from RP models with known connectivity

Synthetic neural activity patterns were obtained by training RP models on real neural recordings as described above and then generating data from these models using Metropolis–Hastings sampling.

## Acknowledgements

We thank Adam Haber, Tal Tamir, Udi Karpas, and the rest of the Schneidman lab members for discussions, comments, and ideas. This work was supported by Simons Collaboration on the Global Brain grant 542997, Israel Science Foundation grant 137628, The Deutsche Forschungsgemeinschaft (DFG, German Research Foundation) - Project-ID 454648639 - SFB 1528, Israeli Council for Higher Education/Weizmann Data Science Research Center, Martin Kushner Schnur, and Mr. & Mrs. Lawrence Feis. ES is the incumbent of the Joseph and Bessie Feinberg Chair. This research was also supported in part by grants NSF PHY-1748958 and PHY-2309135 and the Gordon and Betty Moore Foundation Grant No. 2919.02 to the Kavli Institute for Theoretical Physics (KITP).

## Additional information

### Funding

| Funder | Grant reference number | Author |
|---|---|---|
| Israel Science Foundation | 137628 | Elad Schneidman |
| Simons Foundation | 542997 | Elad Schneidman |
| Gordon and Betty Moore Foundation | 2919.02 | Elad Schneidman |
| DFG, German Research Foundation | SFB 1528 | Elad Schneidman |
| Israeli Council for Higher Education/Weizmann Data Science Research Center | | Elad Schneidman |
| Martin Kushner Schnur, and Mr. & Mrs. Lawrence Feis | Weizmann Institute | Elad Schneidman |
| The Joseph and Bessie Feinberg Chair | Weizmann Institute | Elad Schneidman |

The funders had no role in study design, data collection and interpretation, or the decision to submit the work for publication.

### Author contributions

Jonathan Mayzel, Conceptualization, Software, Validation, Investigation, Visualization, Methodology, Writing – original draft, Writing – review and editing; Elad Schneidman, Conceptualization, Supervision, Funding acquisition, Validation, Investigation, Visualization, Methodology, Writing – original draft, Project administration, Writing – review and editing

### Author ORCIDs

Jonathan Mayzel ![ORCID] https://orcid.org/0009-0009-2237-3880
Elad Schneidman ![ORCID] https://orcid.org/0000-0001-8653-9848

Reviewer #1 (Public review): https://doi.org/10.7554/eLife.96566.3.sa1
Author response https://doi.org/10.7554/eLife.96566.3.sa2

---

## Additional files

### Supplementary files
• MDAR checklist

### Data availability

The current manuscript is a computational study, so no data have been generated for this manuscript. The software that was developed for this work can be found on GitHub, (copy archived at *Schneidman, 2024*).

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
