## [Editor Report · eLife Assessment]

This work is an **important** contribution to the development of a biologically plausible theory of statistical modeling of spiking activity. The authors **convincingly** implemented the statistical inference of input likelihood in a simple neural circuit, demonstrating the relationship between synaptic homeostasis, neural representations, and computational accuracy. This work will be of interest to neuroscientists, both theoretical and experimental, who are exploring how statistical computation is implemented in neural networks.

---

## [Referee Report · Reviewer #1 (Public review)]

Summary

A novel statistical model of neural population activity called the Random Projection model has been recently proposed. Not only is this model accurate, efficient, and scalable, but also is naturally implemented as a shallow neural network. This work proposes a new class of RP model called the reshaped RP model. Inheriting the virtue of the original RP model, the proposed model is more accurate in terms of data fitting and efficient in terms of lower firing rate than the original, as well as compatible with various biological constraints. In particular, the authors have demonstrated that normalizing the total synaptic input in the reshaped model has a homeostatic effect on the firing rates of the neurons, resulting in even more efficient representations with equivalent accuracy. These results suggest that synaptic normalization contributes to synaptic homeostasis as well as efficiency in neural encoding.

Strength

This paper demonstrates that the accuracy and efficiency of the random projection models can be improved by extending the model with reshaped projections. Furthermore, it broadens the applicability of the model under biological constraints of synaptic regularization. It also suggests the advantage of the sparse connectivity structure over the fully connected model for modeling spiking statistics. In summary, this work successfully integrates two different elements, statistical modeling of the spikes and synaptic homeostasis in a single biologically plausible neural network model. The authors logically demonstrate their arguments with clear visual presentations and well-structured text, facilitating an unambiguous understanding for readers.

Discussions

The authors have clearly responded to most of our questions in the revised manuscript and we are happy to recommend publishing the final version of the article as it is. Below, we would like to present some alternative interpretations of the results. These comments are not exclusive with the claims made in the articles; it is rather intended to enhance the understanding of readers by providing additional perspectives.

As summarized above, the main contribution of the work consists of two parts; (1) the reshaped RP model achieved higher performance in explaining the statistics of the spiking activity of cortical neurons with more efficient representations (=lower firing rate), (2) synaptic homeostatic normalization in the reshaped RP model yields even more efficient representations without impairing the fitting performance.

For part (1),

Suppl. Fig. 1B compares reshaped RP models with RP and RP with pruning and replacement (R&P). The better performance of RP with P&R might imply the advantage of pruning over gradient descent in this setting, reflecting the non-convexities of the problem. Alternatively, it might suggest the benefit of the increased number of parameters, since pruning allows the network to explore the broader parameter space during the learning process. This latter view might partially explain the superiority of the reshaped RP model over the original RP model.

It is interesting that the backprop model has higher firing rate than the reshaped model (Fig. 1D). This trend is unchanged when optimization of the neural threshold is also allowed (Supp. Fig. 2A). Since backprop model overperforms reshaped model slightly but robustly, high firing rates in the backprop model might be a genuine feature of the spike statistics.

For part (2),

We note that λ regulates the average firing rate, since maximizing the entropy <-ln p(x)> with a regularization term -λ <Σ_i_ f(x_i_)> results in λ_i_ = λ for all i in the Boltzmann distribution of eq. 2. Suppl. Fig. 2B could be understood as demonstrating this "homeostatic" effect of λ.

Suppl. Fig. 3 could be interpreted as demonstrating the interaction of two different homeostatic mechanisms: one at the firing-rate level (as regulated by λ) and the other at the synaptic level (as regulated by φ). It shows that the range of synaptic constraints where the fitting performance is not impaired differs by the value of λ. For example, if lambda is small (λ = 0.25), synaptic constraint can easily deteriorate the performance; on the other hand, if lambda is large (λ = 4), performance remains unchanged under strict synaptic constraint. Considering that practically we are most interested in the regime where the model performs best (λ = 2.0), an advantageous feature of the homeostatic model is that homeostatic constraint is harmless at λ=2.0 for the wide range of constraints.

---

## [Author Response]

The following is the authors’ response to the original reviews.

**Public Comments:**
(1) We find it interesting that the reshaped model showed decreased firing rates of the projection neurons. We note that maximizing the entropy <-ln p(x)> with a regularizing term -λ <Σ_i_ f(x_i_)>, which reflects the mean firing rate, results in λ_i_ = λ for all i in the Boltzmann distribution. In other words, in addition to the homeostatic effect of synaptic normalization which is shown in Figures 3B-D, setting all λ_i_ = 1 itself might have a homeostatic effect on the firing rates. It would be better if the contribution of these two homeostatic effects be separated. One suggestion is to verify the homeostatic effect of synaptic normalization by changing the value of λ.

This is an interesting question and we, therefore, explored the effects of different values of λ on the performance of unconstrained reshaped RP models and their firing rates. The new supp. Figure 2B presents the results of this exploration: We found that for models with a small set of projections, a high value of λ results in better performance than models with low ones, while for models with a large set of projections we find the opposite relation. The mean firing rates of the projection neurons for models with different values of λ show a clear trend, where higher λ values results in lower mean firing rates.

Thus, these results suggest an interplay between the optimal size of the projection set and the value of λ one should pick. For the population sizes and projection sets we have used here, λ=1 is a good choice, but, for different population sizes or data sets a different value of λ might be better.

Thus, in addition to supp. Figure 2B, we therefore added the following to the main text:

“An additional set of parameters that might affect the Reshaped RP models are the coefficients λ, that weigh each of the projections. Above, we used λ=1 for all projections, here we investigated the effect of the value of λ on the performance of the Reshaped RP models (supp. Figure 2B). We find that for models with a small projection set, high λ values result in better performance than models with low values. We find an opposite relation for models with large number projection sets. (We submit that the performance decrease of Reshaped RP models with high value of λ, as the number of projections grows, is a reflection of the non-convex nature of the Reshaped RP optimization problem).

The mean firing rates of the projection neurons for models with different values of λ show a clear trend, higher λ values results in lower mean firing rates. Thus, we conclude that there is an interplay between the number of projections and the value of λ we should pick. For the sizes of projection sets we have used here, λ=1 is a good choice, but, we note that in general, one should probably seek the appropriate value of λ for different population sizes or data sets.”

In addition, we explored the effect of synaptic normalization on models with different values of λ (supp. Figure 3). We found that homeostatic Reshaped RP models are superior to the non-homeostatic Reshaped RP models: For low values of λ, the homeostatic and Reshaped RP models show similar performance in terms of log-likelihood, whereas the homeostatic models are more efficient. For high values of λ_i_ homeostatic models are not only more efficient but also show better performance. These results indicate that the benefit of the homeostatic model is insensitive to the specific choice of λ.

In addition to supp. Figure 3, we added the following to the main text:

“Exploring the effect of synaptic normalization on models with different values of λ (supp. Figure 3), we find that homeostatic Reshaped RP models are superior to the non-homeostatic Reshaped RP models: For low values of λ, the homeostatic and Reshaped RP models show similar performance in terms of log-likelihood, whereas the homeostatic models are more efficient. Importantly, for high values of λ_i_ homeostatic models are not only more efficient but also show better performance. We conclude that the benefit of the homeostatic model is insensitive to the specific choice of λ.”

(2) As far as we understand, θ_i_ (thresholds of the neurons) are fixed to 1 in the article. Optimizing the neural threshold as well as synaptic weights is a natural procedure (both biologically and engineeringly), and can easily be computed by a similar expression to that of a_ij_ (equation 3). Do the results still hold when changing θ_i_ is allowed as well? For example,a. If θ_i_ becomes larger, the mean firing rates will decrease. Does the backprop model still have higher firing rates than the reshaped model when θ_i_ are also optimized?b. Changing θ_i_ affects the dynamic range of the projection neurons, thus could modify the effect of synaptic constraints. In particular, does it affect the performance of the bounded model (relative to the homeostatic input models)?

We followed the referee’s suggestion, and extended our current analysis, and added threshold optimization to the Reshape and Backpropagation models, which is now shown in supp. Figure 2A. Comparing the performance and properties of these models to ones with fixed thresholds, we found that this addition had a small effect on the performance of the models in terms of their likelihood. (supp. Figure 2A). We further find that backpropagation models with tuned thresholds show lower firing rates compared to backpropagation models with fixed threshold, while reshaped RP models with optimized thresholds show higher firing rates compared to models with fixed threshold. These differences are, again, rather small, and both versions of the reshaped RP models show lower firing rates compared to both versions of the backpropagation models.

In addition to supp. Figure 2A, we added the following to the main text:

“The projections' threshold θ_i_, which is analogous to the spiking threshold of the projection neurons, strongly affects the projections' firing rates. We asked how, in addition to reshaping the coefficients of each projection, we can also change θ_i_ to optimize the reshaped RP and backpropagation models.

We find that this addition has a small effect on the performance of the models in terms of their likelihood (supp. Figure 2A).

We also find that this has a small effect on the firing rates of the projection neurons: backpropagation models with tuned thresholds show lower firing rates compared to backpropagation models with fixed threshold, whereas reshaped RP models with optimized thresholds show higher firing rates compared to models with fixed threshold. Yet, both versions of the reshaped RP models show lower firing rates compared to both versions of the backpropagation models. Given the small effect of tuning threshold on models' performance and their internal properties, we will, henceforth, focus on Reshaped RP models with fixed thresholds.”

(3) In Figure 1, the authors claim that the reshaped RP model outperforms the RP model. This improved performance might be partly because the reshaped RP model has more parameters to be optimized than the RP model. Indeed, let the number of projections N and the in-degree of the projections K, then the RP model and the reshaped RP model have N and KN parameters, respectively. Does the reshaped model still outperform the original one when only (randomly chosen) N weights (out of a_ij_) are allowed to be optimized and the rest is fixed? (or, does it still outperform the original model with the same number of optimized parameters (i.e. N/K neurons)?)

Indeed, the number of tuned parameters in the reshaped RP model is much larger compared to the number of tuned parameters in an RP model with the same projection set size. Yet, we submit that the larger number of tuned parameters is not the reason for the improved performance of the reshaped RP model: Maoz et al [30] have already shown that by optimizing an RP model with a small projection set using the pruning and replacement of projections (P&R), one can reach high accuracy with an almost order of magnitude fewer projections. Thus, we argue that the improved performance stems from the properties of the projections in the model.

Accordingly, we therefore added supp. Figure 2B that shows the performance of P&R sigmoid RP model compared to RP and reshaped RP models. We added the following to the main text:

“Because reshaping may change all the existing synapses of each projection, the number of parameters is the number of projections times the projections in-degree. While this is much larger than the number of parameters that we learn for the RP model (one for each projection), we suggest that the performance of the reshaped models is not a naive result of having more parameters. In particular, we have seen that RP models that use a small set of projections can be very accurate when the projections are optimized using the pruning and replacement process [30] (see also supp. Figure 1B). Thus, it is really the nature of the projections that shapes the performance. Indeed, our results here show that a small fixed connectivity projection set with weight tuning is enough for accurate performance which is on par or better than an RP model with more projections.”

(4) In Figure 2, the authors have demonstrated that the homeostatic synaptic normalization outperforms the bounded model when the allowed synaptic cost is small. One possible hypothesis for explaining this fact is that the optimal solution lies in the region where only a small number of |a_ij_| is large and the rest is near 0. If it is possible to verify this idea by, for example, exhibiting the distribution of a_ij_ after optimization, it would help the readers to better understand the mechanism behind the superiority of the homeostatic input model.

We modified supp. Figure 4 and made the following change in the relevant part in the main text to address the reviewer comment about the distribution of the a_ij_ values:

“Figure 5E shows the mean rotation angle over 100 homeostatic models as a function of synaptic cost -- reflecting that the different forms of homeostatic regulation results in different reshaped projections. We show in Supp. Figure 4C the histogram of the rotation angles of several different homeostatic models, as well as the unconstrained Reshape model.

Analyzing the distribution of the synaptic weights a_ij_ after learning leads to a similar conclusion (supp. Figure 4D): The peak of the histograms is at a_ij_ = 0, implying that during reshaping most synapses are effectively pruned. While the distribution is broader for models with higher synaptic budget, it is asymmetric, showing local maxima at different values of a_ij_.

The diversity of solutions that the different model classes and parameters show imply a form of redundancy in model choice or learning procedure. This reflects a multiplicity of ways to learn or optimize such networks that biology could use to shape or tune neural population codes.“

(5) In Figures 5D and 5E, the authors present how different reshaping constraints result in different learning processes ("rotation"). We find these results quite intriguing, but it would help the readers understand them if there is more explanation or interpretation. For example,a. In the "Reshape - Hom. circuit 4.0" plot (Fig 5D, upper-left), the rotation angle between the two models is almost always the same. This is reasonable since the Homeostatic Circuit model is the least constrained model and could be almost irrelevant to the optimization process. Is there any similar interpretation to the other 3 plots of Figure 5D?

We added a short discussion of this difference to the main text, but do not have a geometric or other intuitive explanation for the nature of these differences.

b. In Figure 5E, is there any intuitive explanation for why the three models take minimum rotation angle at similar global synaptic cost (~0.3)?

We added discussion of this issue to the main text, and the histogram of the rotation angles in Supp Figure 4c shows that they are not identical. But, we don’t have an intuitive explanation for why the mean values are so similar.

**Recommendations for the authors:**
(1) Some claims on the effect of synaptic normalization on the reshaped model sound a little overstated since the presented evidence does not clearly show the improvement of the computational performance (in comparison to the vanilla reshaped model) in terms of maximizing the likelihood of the inputs. Here are some examples of such claims: "Incorporating more biological features and utilizing synaptic normalization in the learning process, results in even more efficient and accurate models." (in Abstract), "Thus, our new scalable, efficient, and highly accurate population code models are not only biologically-plausible but are actually optimized due to their biological features." (in Abstract), or "in our Reshaped RP models, homeostatic plasticity optimizes the performance of network models" (in Discussion).

We changed the wording according to the reviewers’ suggestions.

(2) In equation (1) and the following sentence, θ_j_ (threshold) should be θ_i_.

Fixed

(3) While the authors mention that "reshaping with normalization or without it drives the projection neurons to converge to similar average firing rate values (Figure 3B)", they also claim that "reshaping with normalization implies lower firing rates as well as... (Figure 3E)". These two claims look a little inconsistent to us. Besides, it is not very clear from Figure 3E that the normalization decreases the firing rate (it is clear from Figure 3B, though). How about just deleting "lower firing rates as well as"?

We changed the wording according to the reviewers’ suggestion.

(4) The captions of Figures 4D and 4E should be exchanged.

Fixed

(5) Typo in In Figure 4F: "normalized in-dgreree".

Fixed

(6) In Figure 5D (upper left plot) the choice of "Reshape" and "Bounded3.0" looks a bit weird. Is this the typo of "Hom. cicruit 4.0"?

There is no typo in the figure labels. We discussed the results of figure 5D in our response to point (5) in the public comments list and addressed the upper left panel of figure 5D in the main text.

(7) In the paper, the letter θ represents (1) the threshold of the projection neurons (eq. 1), (2) the "ceiling" value of the bounded model, and (3) the rotation angle of projections (Figure 5). We find this notation a bit confusing and recommend using different notations for different entities.

Thanks for the suggestion, we changed the confusing notations: (1) The threshold of each projection neuron is still θ following the notation of the original RP model formulation [30]. (2) The notation of the “ceiling” value of the bounded model is now ω. (3) The rotation angle of the projections during reshape is now marked by α.